# Lower Spermatozoal *PIWI-LIKE 1* and *2* Transcript Levels Are Significantly Associated with Higher Fertilization Rates in IVF

**DOI:** 10.3390/ijms222111320

**Published:** 2021-10-20

**Authors:** Maria Giebler, Thomas Greither, Diana Handke, Gregor Seliger, Hermann M. Behre

**Affiliations:** Center for Reproductive Medicine and Andrology, Martin Luther University Halle-Wittenberg, Ernst-Grube-Str. 40, 06120 Halle (Saale), Germany; maria.giebler@horizondiscovery.com (M.G.); thomas.greither@medizin.uni-halle.de (T.G.); diana.handke@uk-halle.de (D.H.); gregor.seliger@medizin.uni-halle.de (G.S.)

**Keywords:** male infertility, *PIWI-LIKE* genes, spermatozoa, ART, fertilization rate

## Abstract

The four human *PIWI-LIKE* gene family members *PIWI-LIKE 1–4* play a pivotal role in stem cell maintenance and transposon repression in the human germline. Therefore, dysregulation of these genes negatively influences the genetic stability of the respective germ cell and subsequent development and maturation. Recently, we demonstrated that a lower *PIWI-LIKE 2* mRNA expression in ejaculated spermatozoa is more frequent in men with oligozoospermia. In this study, we analysed how *PIWI-LIKE 1–4* mRNA expression in ejaculated spermatozoa predicts ART outcome. From 160 IVF or ICSI cycles, portions of swim-up spermatozoa used for fertilization were collected, and the total RNA was isolated. *PIWI-LIKE 1–4* mRNA expression was measured by qPCR using TaqMan probes with GAPDH as a reference gene. *PIWI-LIKE 1* and *2* transcript levels in the spermatozoa of the swim-up fraction were positively correlated to each other (r_S_ = 0.78; *p* < 0.001). Moreover, lower *PIWI-LIKE 2* mRNA levels, as well as lower *PIWI-LIKE 1* mRNA levels, in these spermatozoa were positively associated with a fertilization rate ≥ 50% in the respective ART cycles (*p* = 0.02 and *p* = 0.0499, Mann–Whitney *U*-Test). When separately analysing IVF and ICSI cycles, *PIWI-LIKE 1* and *2* transcript levels were only significantly associated to increased fertilization rates in IVF, yet not in ICSI cycles. Spermatozoal *PIWI-LIKE 3* and *4* transcript levels were not significantly associated to fertilization rates in ART cycles. In conclusion, lower levels of spermatozoal *PIWI-LIKE 1* and *2* mRNA levels are positively associated with a higher fertilization rate in IVF cycles.

## 1. Introduction

Infertility is a major health problem worldwide, with an estimated 48.5 million couples suffering from unwilling childlessness [1]. Male infertility contributes to roughly half of these cases, and impacts not only the reproductive function, but is also linked to the overall health of the affected male [2,3]. In most cases of male infertility, disturbances of the spermatogenesis result in the reduction of semen quality, measurable in total sperm count and the proportions of motile sperm and sperm with typical morphology, but may also result in a disturbed DNA condensation or in higher DNA fragmentation. Many couples with infertile men are referred to assisted reproduction therapy (IVF or ICSI), by which the disturbances in the sperm variables are encompassed by the in vitro selection of optimal spermatozoa. However, the selection of human spermatozoa based on criteria such as viability, motility, and morphology does not necessarily reflect in the subsequent sperm fertilizing capacity [4,5,6].

Depending on the technique chosen and the number of oocytes retrieved, the mean fertilization rate, which is a predictor of a subsequent pregnancy, is around 50–70% in ART cycles in Germany. Nevertheless, in some cases, fertilization rates remain below 25%. These decreased fertilization rates may be due to abnormal DNA condensation or even to fragmentation during spermatogenesis. The DNA fragmentation is described by the DNA fragmentation index (DFI), which can be analysed by various different testing systems [7,8]. Several studies have shown an increased DFI to be associated with a significantly lower fertilization rate [9,10]; however, there are also reports indicating no relation between DFI and embryologic outcomes [11,12,13]. Nevertheless, nowadays DFI measurements are performed routinely in many ART clinics to predict the fertilization capacity of the ejaculate [14,15]. Given the fact that these measurements have only a limited prognostic impact, further early markers of a lowered fertilization competence of the ejaculate are warranted as a diagnostic tool, but also as a therapeutic target, which may in future allow the selection of the optimal spermatozoa.

During spermatogenesis, a plethora of molecular factors are engaged in the maturation of a round, metabolic active, differentiation-competent spermatogonia to a mature spermatozoa. It is assumed that around 2000 genes play a role in the spermatogenic differentiation or the processing in the epididymis [16], many of them being associated with male infertility in mouse knock-out models. One striking example is the *PIWI-LIKE* gene family, a subclade of the Argonaute gene family. *PIWI-LIKE* genes are highly conserved among various species, consisting for example of four human family members (*PIWI-LIKE 1–4*; alternatively, *HIWI*, *HILI*, *HIWI3*, and *HIWI2* [17]) or three murine homologs (*Miwi*, *Mili*, *Miwi2* [18]). When generating Miwi-knockout mice, Deng and Lin observed that Miwi^null^ male mice are sterile, while Miwi^null^ female mice remain fertile. They also demonstrated that Miwi^null^ mice exhibited an arrest of spermatogenesis on the round spermatid level, and that no mature sperm is found in the epididymides of these mice [19]. Similarly, Mili^null^ and Miwi2^null^ male mice are also sterile, with Mili^null^ mice exhibiting a spermatogenic block in the early pachytene stage [20] and Miwi2^null^ mice reaching spermatogenic arrest even in the zygotene stage of the first meiosis [21]. Yet, the impact of the human *PIWI-LIKE* genes on male fertility is still under debate.

Previously, we demonstrated that a dysregulated *PIWI-LIKE 1* or *2* expression is associated with altered semen variables [22]. However, currently no data exist on the connection between dysregulated spermatozoal *PIWI-LIKE* gene expression as a surrogate of suboptimal spermatogenesis and the fertilization rates achieved in ART cycles using these spermatozoa. Therefore, the primary aim of this study was to measure *PIWI-LIKE 1–4* mRNA expression in the swim-up fraction of ejaculated spermatozoa used for ART and to correlate these parameters with the corresponding fertilization rate.

## 2. Results

### 2.1. Distribution of PIWI-LIKE Gene Transcripts in the Spermatozoa

Firstly, we quantitatively assessed *PIWI-LIKE* transcript levels in the swim-up fraction and in the cell pellet. *PIWI-LIKE* gene transcripts were measurable in both fractions of the patients’ samples to a different extent. While *PIWI-LIKE 1* expression could be quantified in 96.2% (swim-up fractions) and 98.2% (cell pellets) of the patients’ samples, *PIWI-LIKE 2* was measurable in 74.9% and 75.2% of the samples, respectively. *PIWI-LIKE 4* was only detectable in around one third of the patients’ samples (32.7% and 36.4%, respectively), while *PIWI-LIKE 3* was more frequently detected in the cell pellet fraction (39.4% of the samples) than in the swim-up fraction (17.5%).

Comparing the transcript levels of *PIWI-LIKE 1*–*4* in the different fractions, we observed a significantly higher expression of *PIWI-LIKE 1* mRNA in the cell pellet than in the swim-up phase (median expression 0.00133 in the swim-up fraction vs. 0.00399 in the cell pellet, *p* < 0.001; Mann–Whitney *U* test, see Table 1). On the other hand, *PIWI-LIKE 2* expression was not significantly different between the spermatozoa of both fractions (see Table 1).

In bivariate correlation analyses (according to Spearman-Rho), the mRNA expression of *PIWI-LIKE 1* (r_S_ = 0.57; *p* < 0.001), *PIWI-LIKE 2* (r_S_ = 0.70; *p* < 0.001), *PIWI-LIKE 3* (r_S_ = 0.22; *p* = 0.006), and *PIWI-LIKE 4* (r_S_ = 0.37; *p* < 0.001) in the spermatozoa of the swim-up fraction and the cell pellet correlated to each other highly significantly. Furthermore, the *PIWI-LIKE 1* and *2* transcript levels were highly significant, associated in the spermatozoa of the swim-up, as well as of the cell fraction (r_S_ = 0.78 and r_S_ = 0.49, respectively; *p* < 0.001). Additionally, *PIWI-LIKE 4* transcript levels in spermatozoa of the swim-up fraction was inversely correlated to the *PIWI-LIKE 1* and *2* levels (r_S_ = −0.25 and r_S_ = −0.28, respectively, *p* < 0.001).

### 2.2. Lower PIWI-LIKE 1 and 2 mRNA Expression Is Associated with Higher Fertilization Rates in ART

We compared the *PIWI-LIKE 1*–*4* transcript levels in the spermatozoa of the swim-up fraction used for ART depending on the fertilization rates in the respective cycle. The median fertilization rate of 50% was chosen as a cut-off. In the swim-up spermatozoa fraction of patients with fertilization rates ≥ 50%, a significantly lower *PIWI-LIKE 1* or lower *PIWI-LIKE 2* mRNA level was observed (*p* = 0.0499 and 0.017, respectively, Mann–Whitney *U*-Test, see Figure 1a,b).

Furthermore, in cycles with ≥ 8 inseminated oocytes, the correlation between a lower *PIWI-LIKE 1* or *2* mRNA expression and a fertilization rate ≥50% was even more pronounced (*p* = 0.019 or *p* = 0.009, respectively, Mann–Whitney *U*-test). Spermatozoal *PIWI-LIKE 3* and *PIWI-LIKE 4* mRNA levels did not show any correlations to fertilization rate.

### 2.3. PIWI-LIKE 1 and 2 mRNA Expression Is a Predictor for Fertilization in IVF, but Not ICSI Cycles

When subdividing the whole study cohort according to the ART treatment performed, only in the IVF group a lowered *PIWI-LIKE 1* or *2* mRNA level was significantly associated with a higher fertilization success (*p* = 0.002 and *p* = 0.019, respectively, Mann–Whitney *U*-test; see Figure 2a,b). In comparison, in the ICSI group, no significant association between the spermatozoal *PIWI-LIKE* mRNA expression and the fertilization results was seen (see Figure 2c,d). Of note, only four (7.0%) of male patients in the IVF group exhibited mild impairments semen values, while 70 (72.9%) of the patients in the ICSI group showed abnormal sperm values.

In ROC analyses, both *PIWI-LIKE 1* and *2* transcript levels were fair predictors of a higher fertilization rate in IVF cycles (see Figure 3a). Spermatozoal *PIWI-LIKE 1* mRNA levels exhibited an AUC of 0.72 (asymptotic *p* = 0.002), while *PIWI-LIKE 2* mRNA expression had an AUC of 0.67 (asymptotic *p* = 0.021).

Based on the ROC analyses, we calculated a Youden optimized cut-off value for the spermatozoal *PIWI-LIKE 1* and *PIWI-LIKE 2* transcript levels. Below a spermatozoal *PIWI-LIKE 1* mRNA level of 0.0001 (ΔCq), the mean fertilization rate was significantly higher than above this cut-off (53.9% vs. 36.0%; *p* = 0.011, Mann–Whitney *U*-test). Regarding spermatozoal *PIWI-LIKE 2* mRNA levels, below 0.00001 (ΔCq), the mean fertilization rate was significantly increased (51.5% vs. 35.5%; *p* = 0.022; Mann–Whitney *U*-Test, see Figure 3b).

## 3. Discussion

In this study, we observed, for the first time, an association between low spermatozoal *PIWI-LIKE 1* and *2* transcript level and a higher fertilization rate in IVF. As *PIWI-LIKE* gene expression was directly assessed in the motile spermatozoa from the swim-up fraction used in the respective ART cycle, an influence of an altered expression of the *PIWI-LIKE* genes during spermatogenesis and reproductive success may be hypothesized. Furthermore, the results of this study may pave the road to the inclusion of *PIWI-LIKE* gene expression as an additional marker for the clinical decision between IVF or ICSI as treatment of choice, if confirmed in larger independent studies.

We were able to quantify the *PIWI-LIKE* mRNA expressions of the four human family members in the spermatozoa of the swim-up and the primarily immotile cell fraction to a different extent. While *PIWI-LIKE 1* and *2* mRNA were detectable in over 95% and in around 75% of the samples, *PIWI-LIKE 3* and *4* transcripts were present in a fewer number of samples. This is concordant with our previous study, where we could detect *PIWI-LIKE 1* in all studied samples, and *PIWI-LIKE 2* in around 50% of the samples, while *PIWI-LIKE 3* and *PIWI-LIKE 4* were only measurable in less than 20% of the samples [22]. Furthermore, *PIWI-LIKE 4* expression is inversely correlated to the expression of *PIWI-LIKE 1/2*. These findings are in concordance with the known expression sequence of PIWI proteins in the mammalian system, with *PIWI-LIKE 4* being expressed in early stages, while *PIWI-LIKE 2* is expressed over the main course of spermatogenesis, and finally being replaced by *PIWI-LIKE 1* expression in the mid-pachytene stage (reviewed in [23]). In concordance with our results, Hempfling and colleagues demonstrated the mRNA expression of *PIWI-LIKE 1* and *2* in the vast majority of the testis tissue samples displaying normal or impaired spermatogenesis [24].

Although decreased *PIWI-LIKE 2* expression was associated to impaired semen parameters [22] in our previous study, our current study shows a relation between lower *PIWI-LIKE 2* expression and higher fertilization success. The contradictory results regarding *PIWI-LIKE 2* and sperm parameters compared to our first study could be explained by the assessment of spermatozoa from native ejaculate in the initial study versus the analysis of spermatozoa solely from the swim-up phase in the present study. However, both studies strongly suggest that *PIWI-LIKE* gene expression serves as a predictor for semen quality.

We observed a significant association between a lower *PIWI-LIKE 1* or *2* transcript expression in the spermatozoa used for IVF and increased fertilization rates. To the best of our knowledge, this is the first time that *PIWI-LIKE* gene expression could be linked to fertilization success. Cui and colleagues demonstrated that the increased spermatozoal expressions of the piwi-interacting RNAs piR-31704, piR-39888, and piR-40349 are linked with a fertilization success >70% in 186 ICSI cycles [25]. A study from Denomme and colleagues showed that a group of ten miRNAs were identified as significantly altered in association with poor blastocyst development, with corresponding alterations to target genes involved in embryonic genome activation, blastocyst implantation, and DNA methylation [26]. The complex interplay of the PIWIL/piRNA machinery with gametogenesis and fertilization is still not elucidated yet.

The impact of the PIWI-LIKE proteins on human spermatogenesis is still unclear. In 2017, Gou and colleagues described several heterozygous mutations in the D-box region of *HIWI/PIWI-LIKE 1* gene of three out of 413 patients with idiopathic azoospermia, thereby impairing the ubiquitination of PIWI-LIKE 1 [27]. Additionally, the authors demonstrated in mice that the resulting stabilization and accumulation of PIWI-LIKE 1 induces abnormal chromatin compaction due to an impaired histone–protamine exchange and thus leading to male sterility [27]. However, recently Oud and colleagues showed in a larger study cohort of 2740 patients with azoospermia or severe oligozoospermia, that HIWI D-box variations are less frequent than proposed by Gou and colleagues, and the that loss of function of PIWI-LIKE 1 is unlikely to cause fertility impairments when haploin-sufficient [28]. In our study, we did not perform mutational screenings on the *PIWI-LIKE* genes, therefore statements about the relationship of mutated *PIWI-LIKE* genes and male infertility cannot be drawn from our experimental setting.

Earlier studies showed that spermatozoal RNAs are also delivered to the oocyte and potentially contribute to early embryo development [29,30]. Although murine PIWI-LIKE 2 was detected alongside other nuage structure proteins in primordial follicular oocytes [31], less is known about the role of *PIWI-LIKE* genes in human oocytes. It can be hypothesized, that *PIWI-LIKE* transcript delivery via the spermatozoon may assist in retrotransposon control during early embryonic development [32].

A limitation of our study is that we do not have insight into all maternal factors involved in the fertilization process. Fertilization is a multifactorial process that involves maternal and paternal interplay. Success and failure of oocyte fertilization cannot be referred solely to one single gene family in men but remain multifactorial.

Furthermore, the evaluation of mRNA in spermatozoa as a predictor for male fertilization capacity is still experimental. The correlation between RT-qPCR data and fertilization success does not exactly reflect the mRNA content of the single spermatozoon that induces the fertilization of one particular oocyte.

## 4. Materials and Methods

### 4.1. Ethical Approval

The ethics committee of the Medical Faculty of the Martin Luther University Halle-Wittenberg approved the study. All patients gave written informed consent.

### 4.2. Study Population

Patients were included in this study if they were 18–50 years old and signed a written informed consent. Exclusion criteria were azoospermia, known genetic causes of infertility such as cystic fibrosis or Y chromosomal microdeletions, and testosterone or other drug abuse. ART cycles with at least one oocyte inseminated were included in this study. A total of 130 patients were included in this study, with 67 exhibiting normal semen values, while, in 63 patients, one or more sperm parameters were below the reference ranges defined in the WHO laboratory manual for the examination and processing of human semen, 5th edition. Of the patients with abnormal sperm values, 26 patients exhibited an oligozoospermia, 7 asthenozoospermia, and 2 teratozoospermia. Eleven patients had an oligoteratozoospermia, 7 patients oligoasthenozoospermia and 10 patients oligoasthenoteratozoospermia. For more information on semen values see Table 2.

In total, 160 ART cycles were performed at the Center for Reproductive Medicine and Andrology of the University Hospital Halle (Saale), either by in vitro fertilization (IVF; *n* = 64) or by intracytoplasmatic sperm injection (ICSI; *n* = 96). In 19 treatment cycles (11.9%), no fertilization was achieved (IVF: 11 cyles; 17.2%; ICSI: 8 cycles; 8.3%). In median, there were 9 oocytes retrieved per cycle, of which 8 could be inseminated. Median fertilization rate was 50%, and median clinical pregnancy rate was 26.3%. For detailed cycle information, see Table 3.

### 4.3. Specimen Preparation and RNA Isolation

After collection, the ejaculate was stored 30 min at 37 °C for liquefaction. Sperm concentration and motility were assessed as described in the WHO manual, 5th edition, with at least 2 × 200 spermatozoa counted for each parameter. After ejaculate analysis, the specimens were centrifuged (1000× *g*, 10 min) on a density gradient (Sil Select, FertiPro, Berneem, Belgium) to exclude cell debris and non-spermatozoal cells. The cell pellet was washed with Gamete Buffer (Cook Medical, Bloomington, IN, USA), subsequently overlaid with Gamete Buffer and incubated for 30 min at 37 °C. ART (either IVF or ICSI) was performed with spermatozoa from the swim-up fraction. Directly after the insemination, the remaining swim-up fraction and cell pellet were separated, washed with sterile PBS (Invitrogen, Carlsbad, CA, USA), and immediately stored at −80 °C until RNA isolation.

Total RNA was isolated using the RNeasy Micro Kit (Qiagen, Hilden, Germany). Briefly, cell pellets of either the swim-up or the cellular fraction were lysed with RLT buffer + 1% ß-mercaptoethanol (Sigma-Aldrich, St. Louis, MO, USA), and RNA was precipitated with 70% ethanol (Sigma-Aldrich, St. Louis, MO, USA). The total RNA was bound to a silica spin column and treated with DNase I (Sigma-Aldrich, St. Louis, MO, USA) for 10 min to remove residual DNA contaminations. After three rounds of washing with the appropriate buffers, the RNA was eluted in 14 μL RNAse-free water. RNA concentration and purity were assessed by absorption spectrometry (Eppendorf, Hamburg, Germany).

### 4.4. cDNA Synthesis and Quantitative Real-Time PCR

Spermatozoal RNA was transcribed to cDNA with the RevertAid First Strand cDNA Synthesis Kit (Thermo Fisher, Waltham, MA, USA) according to the manufacturers protocol using a mixture of random hexamer primers and OligodT primers. Quantitative PCR of the *PIWI-LIKE* genes was applied with TaqMan Primer (Applied biosystems; Waltham, MA, USA) on a MyIQ-Cycler (BioRad, Hercules, CA, USA). *GAPDH* expression was used as internal reference, and ΔCq values were calculated according to Livak and Schmittgen [33].

### 4.5. Statistical Analyses

Statistical analyses were performed using SPSS (Version 25, SPSS Inc., Chicago, IL, USA). Bivariate correlation analyses according to Spearman-Rho and non-parametric tests (Mann–Whitney *U*-Test) were conducted for the evaluation of the association between human *PIWI-LIKE 1*–*4* mRNA expression, ejaculate variables, and fertilization rates. *p* < 0.05 was considered significant.

## 5. Conclusions

In conclusion, when analyzing the spermatozoal *PIWI-LIKE* gene transcript levels in the swim-up phase of sperm used for ART, lower *PIWI-LIKE 1* and *2* transcript levels were associated with increased fertilization rates in IVF cycles. Based on these results, it could be speculated that spermatozoal *PIWI-LIKE 1* and *2* might be a surrogate marker for proper sperm development and fertilization capacity of sperm used for IVF treatment, and that, therefore, patients with higher *PIWI-LIKE 1* or *2* transcript levels in swim-up sperm should be offered ICSI despite having relatively good classical sperm values.

## Figures and Tables

**Figure 1 ijms-22-11320-f001:**
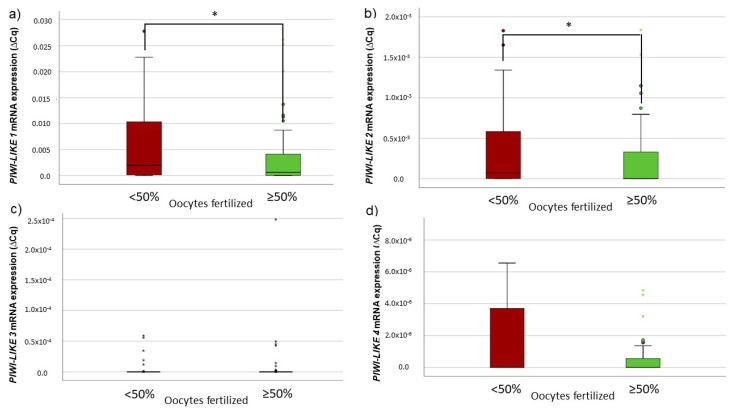
Association between fertilization success and (**a**) *PIWI-LIKE 1*, (**b**) *PIWI-LIKE 2*, (**c**) *PIWI-LIKE 3*, or (**d**) *PIWI-LIKE 4* spermatozoal mRNA levels in 160 ART cycles. * *p* < 0.05.

**Figure 2 ijms-22-11320-f002:**
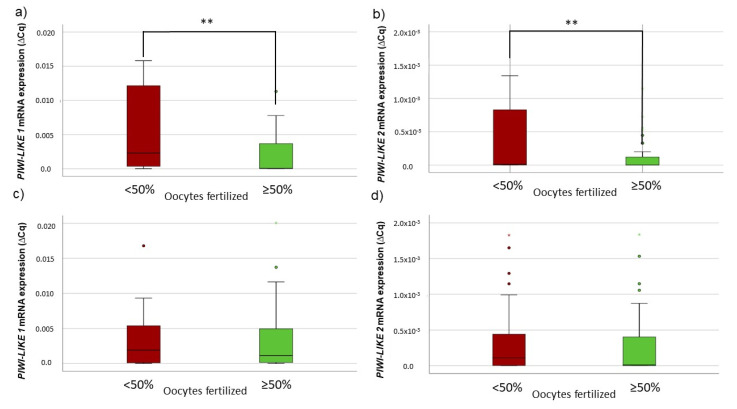
Association between fertilization success and spermatozoal *PIWI-LIKE 1* spermatozoal mRNA levels in (**a**) IVF or (**c**) ICSI cycles and association between fertilization success and spermatozoal *PIWI-LIKE 2* mRNA levels in (**b**) IVF or (**d**) ICSI cycles. ** *p* <0.01.

**Figure 3 ijms-22-11320-f003:**
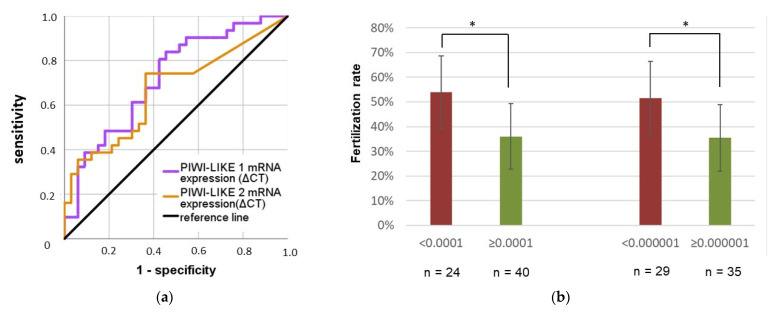
Potential of *PIWI-LIKE 1* and *2* spermatozoal mRNA level to distinguish between high and low-fertilization patient groups applying (**a**) ROC analyses and (**b**) different cut-off rates for *PIWI-LIKE 1* and *2*. * *p* < 0.05.

**Table 1 ijms-22-11320-t001:** Distribution of *PIWI-LIKE* gene transcripts in spermatozoa from the swim-up and the cell pellet fraction.

		Median Expression (ΔCq)	*p* (MW*U*)	*n*
*PIWI-LIKE 1*	swim-up fraction	0.00113	<0.001	160
	cell pellet	0.00399		
*PIWI-LIKE 2*	swim-up fraction	0.00001	n.s.	160
	cell pellet	0.000012		
*PIWI-LIKE 3*	swim-up fraction	0.00	<0.001	160
	cell pellet	0.00		
*PIWI-LIKE 4*	swim-up fraction	0.00	n.s.	160
	cell pellet	0.00		

Abbreviation: MW*U*–Mann–Whitney *U*-test.

**Table 2 ijms-22-11320-t002:** Demographical and clinical data on the patients’ cohort. The minimal and maximal values are given in brackets.

Parameter	Male Partner	Female Partner
age (y)	37 (25–52)	34 (26–41)
weight (kg)	88.4 (58–150)	67.1 (43–123)
BMI (kg/m^2^)	26.9 (18.3–41.6)	24.0 (16.5–40.2)
days of abstinence	4 (2–21)	
semen volume (mL)	2.8 (0.1–10.4)	
sperm count (total sperm/ejaculate)	70.1 (0.1–717.6)	
progressive motility (%)	50 % (0–90%)	
% morphologically normal spermatozoa	8% (0–24%)	

**Table 3 ijms-22-11320-t003:** Clinical data on the ART cycles. The minimal and maximal values are given in brackets.

	Total (*n* = 160)	IVF (*n* = 64)	ICSI (*n* = 96)
Cycles with male indication for ART	26	0	26
Cycles with female indication for ART	61	56	5
Cycles with both partner indication for ART	73	8	65
Oocytes retrieved (n)	9 (1–36)	10 (1–25)	8 (1–36)
Inseminated (n)	8 (1–25)	10 (1–25)	6 (1–19)
Fertilized (n)	4 (0–14)	4 (0–14)	3 (0–10)
Fertilization rate (%)	50 (0–100)	50 (0–100)	58 (0–100)
Transferred embryos (n)	2 (0–3)	2 (0–3)	2 (0–3)
Clinical pregnancy rate (n)	26.3	26.6	26.0

## Data Availability

Original data are available from the authors by reasonable behalf.

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
