# Peer review of "Lower Spermatozoal PIWI-LIKE 1 and 2 Transcript Levels Are Significantly Associated with Higher Fertilization Rates in IVF"

_ijms, 2021, doi:10.3390/ijms222111320_

Round 1

Reviewer 1 Report

The manuscript titled “Lower spermatozoal PIWI-LIKE 1 and 2 transcript levels are sig- 2 nificantly associated with higher fertilization rates in IVF”.

The authors focus the attention on the need find a genetics marker for discriminate the sperm in relation of its fertilizing capacity. In particular was evaluated the PIWI-LIKE 1-4 mRNA expression in the swim-up fraction of ejaculated spermatozoa used for ART to correlate these parameters with the corresponding fertilization rate.

Thanks to this technique the authors concluded that, becouse was observed that low espression level of PIWI-LIKE 1-4 mRNA are present in sperm selected with major fertilizing capacity this can be speculated that PIWI-LIKE 1 and 2 might be a marker for proper sperm development and fertilization capacity of the ejaculate. Therefore patients with higher PIWI-LIKE 1 or 2 transcript levels in swim-up sperm should be offered ICSI despite relatively good classical sperm values.

It is an interesting paper, however, there are many points that need to be addressed.

Major

1) Abstract section Line 19-26. Describe better please these results.

2) Introduction section: Line 30-33 Explain the impact that have the male infertility on the overall health man, what do you mean?

3) Introduction section: Line 33-35. The semen quality alterations due at altered spermatogenesis can depend also of sperm DNA fragmentation. Introduce this concept please.

4) Introduction section. Line 59-65. Explain better this concept please. The phrase is confused.

5) Results section. Line 99. Explain better this results please.

6)Results section. Line 103-104 Explain better this comparation.

7) Results section Line 112. What do you mean with "follicular puncture " explain better this procedure and this concept.

8)Results section Line 119-124. How do you explain this results?

9) Pag 5 Figure 3 (b). Show better the figure in relation at the numbers.

10) Discussion section. Line 148. The authors identify low spermatozoal PIWI-LIKE 1 and 2 resposible of the higher fertilization potential in motile spermatozoa from swim- up fraction, but it's not clear becouse this carachteristics is osservable only in IVF procedure and not in ICSI , explain the probably cause.

11) Material and Methods section . Introduce and describe in the text, please the inclusion and exclusion criteria patients and the their partners. Furthermore a control group is absent.

12) Material and Methods section: Describe please the semen quality of the patients with parameters that are below the reference ranges of the WHO guidelines.

13) Materials and Methods: Describe the criteria followed for to direct the patients at IVF or ICSI procedure.

14) Results section. Line 220-223. Explain in the text the  results related to table 3.

15)Table 3 . Make again the table 3, it is confused.

16) Table 3 : Specify please the characteristics of the semen quality in the cycles with male indication for ART.

17) Conclusion section: Line 259-263: in Material and Methods section the autors write: "In comparison, in the ICSI group no significant association be

tween the spermatozoal PIWI-LIKE mRNA expression and the fertilization results was

seen " this results is in contradiction with the Conclusion Line 259-262: The authors write"Based on these results, it could be speculated that spermatozoal PIWI-LIKE 1 and 2 might be a surrogate marker for proper sperm development and fertilization capacity of the ejaculate, and that therefore patients with higher PIWI-LIKE 1 or 2 transcript levels in swim-up sperm should be offered ICSI despite relatively good classical sperm values".

However, often, the ICSI procedure is used in patients  with altered semen quality as criptozoospermia or severe oligo-astenoospermia where the  procedure ART don't contemplate the swim up for sperm selection. Give information ad results on this aspect please. Furthermore, where is applicable this method in the clinic, on which category of patients, becouse as it is a very expensive technique.

Minor

1) Introduction section: Line 33-35. Replace" morphologically regular sperm cells" with" sperm with typical morphology".

2) Introduction section. Line 45 "The DNA fragmentation is evalueted by DFI and analyzed by many tests".

3) Introduction section Line 56: Replace please "metabolically inactive spermatozoon with compacted DNA" with "mature sperm".

4) Discussion section. Line 167 Correct the reference please.

5) Reference section. To correct the reference please.

Author Response

 We want to thank the reviewer for the helpful revisions, which – to our opinion – greatly helped to improve the manuscript. Thank you for your time and insightful remarks. Revisions according to the questions raised are detailed as follows:

Major

Q1) Abstract section Line 19-26. Describe better please these results.

A1) We apologize for the unclear presentation of our data in the abstract section. L.19 – 26 have been restructured and read now as follows:

“From 160 IVF or ICSI cycles, portions of swim-up spermatozoa used for fertilization were collected and total RNA was isolated. PIWI-LIKE 1-4 mRNA expression was measured by qPCR using TaqMan probes with GAPDH as reference gene. PIWI-LIKE 1 and 2 transcript levels in the spermatozoa of the swim-up fraction were positively correlated to each other (rS = 0.78; p<0.001). Moreover, lower PIWI-LIKE 2 mRNA levels as well as lower PIWI-LIKE 1 levels in these spermatozoa were positively associated with fertilization rate ≥ 50% in the respective ART cycles (p = 0.02 and p = 0.0499, Mann-Whitney U-Test). When separately analysing IVF and ICSI cycles, PIWI-LIKE 1 and 2 transcript levels were only significantly associated to increased fertilization rates in IVF, yet not in ICSI cycles. Spermatozoal PIWI-LIKE 3 and 4 transcript levels were not significantly associated to fertilization rates in ART cycles. In conclusion, lower levels of spermatozoal PIWI-LIKE 1 and 2 mRNA levels are positively associated with a higher fertilization rate in IVF cycles.”

Q2) Introduction section: Line 30-33 Explain the impact that have the male infertility on the overall health man, what do you mean?

A2) We tried to clarify the respective sentence, as our previous statement was too simplified. In short: the adherent references show from large register studies, that male infertility is associated with an increased risk of mortality.

Q3) Introduction section: Line 33-35. The semen quality alterations due at altered spermatogenesis can depend also of sperm DNA fragmentation. Introduce this concept please.

 A3) According to the reviewers suggestion we mentioned DNA fragmentation in l. 43. The sentence now states: “In most cases of male infertility, disturbances of the spermatogenesis result in the reduction of semen quality, measurable in total sperm count and the proportions of motile and sperm with typical morphology, but may also result in disturbed DNA condensation or higher DNA fragmentation.” The concept is than revisited in l. 52-54 following.

Q4) Introduction section. Line 59-65. Explain better this concept please. The phrase is confused.

A4) We changed the according passage as follows: “When generating Miwi-knockout mice, Deng and Lin observed that Miwinull male mice are sterile, while Miwinull female mice remain fertile. They also demonstrated that Miwinull mice exhibited a spermatogenesis arrest on the round sperm spermatid level, and that no mature sperm is found in the epidydimides of these mice [19]. –Similarly, Milinull and Miwi2null male mice are also sterile, with Milinull mice exhibiting a spermatogenic block in the early pachytene stage [20] and Miwi2null mice reaching spermatogenic arrest even in the zygotene stage of the first meiosis [ 21]. Yet, the impact of the human PIWI-LIKE genes on male fertility is still under debate.”

Q5) Results section. Line 99. Explain better this results please.

 A5) Our observation, that a low expression of PIWI-LIKE 4 transcripts and an increased expression of PIWI-LIKE 1 and 2 transcripts in the analyzed swim-up samples are significantly associated, in in line with the already established expression patterns of the different PIWI-LIKE genes, where PIWI-LIKE 4 is a marker for primordial germ cells and spermatogonia, while PIWI-LIKE 1 and 2 are initially expressed with the start of spermatogenesis. Therefore, one may speculate that the inverse association we saw in our analyses reflects these expression patterns. However, as this observation is only based on correlation and can not substantiate causality, we would not like to interpret this observation in the manuscript.

Q6)Results section. Line 103-104 Explain better this comparation.

A6)  We rearranged the sentence as follows: “We compared the PIWI-LIKE 1-4 transcript levels in the spermatozoa of the swim-up fraction used for ART depending on the fertilization rates in the respective cycle. The median fertilization rate of 50% was chosen as cut-off.

Q7) Results section Line 112. What do you mean with "follicular puncture " explain better this procedure and this concept.

A7) A prerequisite of ART treatments by IVF or ICSI is the extraction of the patient’s oocytes from the ovaries for the insemination with the partner’s spermatozoa in vitro. This is achieved by ultra-sound guided puncture of the individual follicles. However, due to the reviewers comment, we decided to re-formulate the sentence, as the procedure of oocyte retrieval is not relevant in the context of our analyses. Therefore, the sentence reads now as follows: “Furthermore, in cycles with ≥ 8 inseminated oocytes the correlation between a lower PIWI-LIKE 1 or 2 mRNA expression and a fertilization rate ≥50% was even more pronounced (p = 0.019 or p = 0.009, respectively, Mann-Whitney U test).”

Q8) Results section Line 119-124. How do you explain this results?

A8)  Our observation, that spermatozoal PIWI-LIKE 1 and 2 transcript levels are only associated with fertilization rates in IVF, yet not in ICSI cycles may be due to the fact, that in ICSI the spermatozoon used for the insemination of the oocyte is picked intentionally by the embryologist, based on the morphological appearance and personal experience. Therefore, in ICSI this manual selection of the inseminated spermatozoon may bypass detrimental effects of a higher proportion of spermatozoa not properly developed.

Q9) Pag 5 Figure 3 (b). Show better the figure in relation at the numbers.

A9)  We tried to enhance the quality of the figure, so the numbers are better readable.

Q10) Discussion section. Line 148. The authors identify low spermatozoal PIWI-LIKE 1 and 2 resposible of the higher fertilization potential in motile spermatozoa from swim- up fraction, but it's not clear becouse this carachteristics is osservable only in IVF procedure and not in ICSI , explain the probably cause.

A10)  We agree with the reviewer, that in our study we can only show correlations, while the mechanistic basis of these observations remain for future studies. Additionally, we also see the point made by the reviewer, that sperm motility is a parameter only of interest in IVF, while in ICSI in theory it is not essential if the injected spermatozoon is motile. However, in all cases we analyzed in this study, there were >90% motile spermatozoa in the swim-up fraction, and it is good clinical practice in our center to only microinject progressively motile and morphologically normal spermatozoa via ICSI. Therefore, it is highly likely to assume that every fertilization achieved in our study was initiated by a motile spermatozoon.

Q11) Material and Methods section . Introduce and describe in the text, please the inclusion and exclusion criteria patients and the their partners. Furthermore a control group is absent.

A11) Inclusion and exclusion criteria were as follows: Patients were included in this study, if they were 18 – 50 years old and signed a written informed consent. Exclusion criteria were azoospermia, known genetic causes of infertility such as cystic fibrosis or Y chromosomal microdeletions and testosterone or other drug abuse. ART cycles with at least one oocyte inseminated were included in this study.”

Given the absence of a control group: as in this study no intervention besides ART was used, a proper control group would be couples with natural conception. However, as the study design was primarily based on the analysis of the spermatozoa from the ejaculate definitively generating the fertilization of one or several oocytes (and in the long run a conception), the applicability in natural conceiving couples would be difficult or even impossible.

Q12) Material and Methods section: Describe please the semen quality of the patients with parameters that are below the reference ranges of the WHO guidelines.

A12) We specified the diagnoses in l. 252-256: “Of the patients with abnormal sperm values, 26 patients exhibited an oligozoospermia, 7 asthenozoospermia, and 2 teratozoospermia. 11 patients had an oligoteratozoospermia, 7 patients oligoasthenozoospermia and 10 patients oligoasthenoteratozoospermia

Q13) Materials and Methods: Describe the criteria followed for to direct the patients at IVF or ICSI procedure.

A13) According to German legislation, couples with severe impairment of male fertility or previous fertilization failure are treated by ICSI.  Couples with normal or mildly impaired semen values and/or factors of female infertility are treated by IVF.

Q14) Results section. Line 220-223. Explain in the text the  results related to table 3.

 A14) We linked l. 220 – 223 to table 3 by adding the sentence: “In median, there were 9 oocytes retrieved per cycle, of which 8 could be inseminated. Median fertilization rate was 50%, and median clinical pregnancy rate was 26.3%.”

Q15) Table 3 . Make again the table 3, it is confused.

 A15) We tried to enhance the clarity of table 3.

Q16) Table 3 : Specify please the characteristics of the semen quality in the cycles with male indication for ART.

A16)  Cycles with male indication for ART consisted for patients with impaired semen values. Of the 26 cycles included, there were 3 patients with asthenozoospermia, 1 patient with teratozoospermia, 5 patients with oligoteratozoospermia, 7 patients with oligoasthenozoospermia and 10 patients with OAT syndrome.

Q17) Conclusion section: Line 259-263: in Material and Methods section the autors write: "In comparison, in the ICSI group no significant association between the spermatozoal PIWI-LIKE mRNA expression and the fertilization results was seen " this results is in contradiction with the Conclusion Line 259-262: The authors write "Based on these results, it could be speculated that spermatozoal PIWI-LIKE 1 and 2 might be a surrogate marker for proper sperm development and fertilization capacity of the ejaculate, and that therefore patients with higher PIWI-LIKE 1 or 2 transcript levels in swim-up sperm should be offered ICSI despite relatively good classical sperm values".

A17) We specified the conclusion line as follows: ”Based on these results, it could be speculated that spermatozoal PIWI-LIKE 1 and 2 might be a surrogate marker for proper sperm development and fertilization capacity of sperm used for IVF treatment, and that therefore patients with higher PIWI-LIKE 1 or 2 transcript levels in swim-up sperm should be offered ICSI despite relatively good classical sperm values.”

Q18) However, often, the ICSI procedure is used in patients  with altered semen quality as criptozoospermia or severe oligo-astenoospermia where the  procedure ART don't contemplate the swim up for sperm selection. Give information ad results on this aspect please. Furthermore, where is applicable this method in the clinic, on which category of patients, becouse as it is a very expensive technique. 

A18) As stated above, there were 10 patients with OAT syndrome included in this study, 8 of them exhibiting sperm values below 1.0 mio/ml. Patients with cryptozoospermia were not included in this study, as with our approach it was not possible to isolate RNA from single spermatozoa and analyze it via qPCR. However, when excluding these eight cases, no significant association between PIWI-LIKE 1 or 2 transcript levels and fertilization rate occurred. The same was true when excluding all cycles with sperm concentrations between 5.0 mio/ml. Therefore, we hypothesize that the spermatozoa selection in ICSI may bypass natural selection processes.

Finally, we agree with the reviewer, that ICSI is an expensive technique. However, to our opinion it is in the best interest of the patients to offer them the technique, which is not only the most cost-effective for them, but also in the sense of personalized medicine the one with the best success rates.

Minor

Q19) Introduction section: Line 33-35. Replace" morphologically regular sperm cells" with" sperm with typical morphology".

 Introduction section. Line 45 "The DNA fragmentation is evalueted by DFI and analyzed by many tests".

 Introduction section Line 56: Replace please "metabolically inactive spermatozoon with compacted DNA" with "mature sperm".

A19) We made the adherent corrections according to your suggestions.

Q20) Discussion section. Line 167 Correct the reference please + Reference section. To correct the reference please.

A20) We thank the reviewer for the correction of this mistake. The reference as well as the reference section were revised according to the correct references order.

Reviewer 2 Report

Comments and Suggestions

Abstract

The authors should start the abstract: Four human family members….

In this section, the PIWI-LIKE 3-4 results must be incorporated

Introduction

It is necessary to add some references in the lines 38-40 and lines 48-40.

The authors must better highlight the importance of PIWI-LIKE genes in male fertility.

Line 70: The semen does not have fertilization capacity (to rewrite)

Results

The authors should compare the levels of PIWI-LIKE mRNA expression between the samples with normal semen values (n=67) and altered semen values (n=63).

It would be necessary to analyze how many normal and altered samples were included in each group (IVF and ICSI), to compare the PIWI-LIKE mRNA expression results between them.

Discussion

Line 171-177: It is necessary to discuss and justify the results on PIWI-LIKE2 expression between swim-up fraction and cell pellet.

The figures are of low quality.

Table 1: Rewrite line 92. This is not correct

Author Response

Firstly, we want to thank the reviewer for the thorough revision of our work as well as the many valuable comments and additional ideas for further analyses. We have included revisions to the manuscript according to the comments as detailed below.

Comments and Suggestions

Q1) Abstract: The authors should start the abstract: Four human family members….; In this section, the PIWI-LIKE 3-4 results must be incorporated

A1) We have revised the abstract section accordingly and incorporated a sentence on the results of the PIWI-LIKE 3 and 4 measurements. 

Q2) Introduction: It is necessary to add some references in the lines 38-40 and lines 48-40.

A2) We inserted references for the adherent sentences. 

Q3) Introduction: The authors must better highlight the importance of PIWI-LIKE genes in male fertility.

A3) We apologize for the unclear presentation of the respective data on the impact of the murine PIWI-LIKE genes on spermatogenesis. We changed the according passage as follows:

When generating Miwi-knockout mice, Deng and Lin observed that Miwinull male mice are sterile, while Miwinull female mice remain fertile. They also demonstrated that Miwinull mice exhibited an arrest of spermatogenesis on the round spermatid level, and that no mature sperm is found in the epididymides of these mice [19]. Similarly, Milinull and Miwi2null male mice are also sterile, with Milinull mice exhibit a spermatogenic block in the early pachytene stage [20] and Miwi2null mice reach spermatogenic arrest even in the zygotene stage of the first meiosis [ 21]. Yet, the impact of the human PIWI-LIKE genes on male fertility is still under debate.

Q4) Introduction: Line 70: The semen does not have fertilization capacity (to rewrite)

A4) We changed the respective sentence as follows: “However, upon now no data exist on the connection between a dysregulated spermatozoal PIWI-LIKE gene expression as surrogate of suboptimal spermatogenesis and the fertilization rates achieved in ART cycles using these spermatozoa.”

Q5) Results: The authors should compare the levels of PIWI-LIKE mRNA expression between the samples with normal semen values (n=67) and altered semen values (n=63).

A5) We thank the reviewer for the suggestion. We performed Mann-Whitney U tests on our data, comparing the PIWI-LIKE 1-4 transcript levels between samples with normal and abnormal semen values. In whole cohort, only PIWI-LIKE 4 expression showed a significant association to the occurrence of abnormal semen values (p = 0.023), mainly based on a higher expression in patients with subnormal ranges of morphological normal spermatozoa (p = 0.005). When analyzing the subcohorts of patients with a IVF or a ICSI treatment independently, in the IVF patients PIWI-LIKE 2 mRNA expression was significantly associated with the occurrence of abnormal sperm values (p = 0.029). However, it should be noted that only 10 (7.0%) of the patients in this group exhibited mild fertility disorders. On the other hand, in the ICSI group no association between semen values and spermatozoal PIWI-LIKE transcript levels could be shown. We added a supplementary figure to illustrate the analyses.

Q6) Results: It would be necessary to analyze how many normal and altered samples were included in each group (IVF and ICSI), to compare the PIWI-LIKE mRNA expression results between them.

A6) As IVF is a technique performed at our center with male patients, which only  show mild fertility issues, there are only 4 (7.0%) patients with abnormal semen parameters in this group. On the other hand, ICSI treatment is the method of choice for patients with severe male fertility impairments, therefore 70 patients (72.9%) exhibited abnormal semen values. We added these information in l.143-145: “Of note, only 4 (7.0%) of male patients in the IVF group exhibited mild fertility impairments semen values, while 70 (72.9%) of the patients in the ICSI group showed abnormal sperm values.”

Q7) Discussion: Line 171-177: It is necessary to discuss and justify the results on PIWI-LIKE2 expression between swim-up fraction and cell pellet.

Q8) The figures are of low quality.

A8) We tried to enhance the quality of the figures to hopefully meet the reviewers expectations.

Q9) Table 1: Rewrite line 92. This is not correct

A9) We tried to adjust l. 92 to the previous paragraph, and corrected inconsistencies.

Reviewer 3 Report

The study focuses on the expression levels of PIWI-LIKE 1 and 2 which could become novel markers for the prediction of a successful IVF or ICSI. In this sense, the manuscript highlights a very up-to-date topic that could have future implications for practical andrology.

The study is compact and logical, the content reads well. I would suggest a small linguistic revision since there are few typos and grammar errors in the manuscript. The references selected for the background and discussion are appropriate.

My questions and/or queries are primarily concenrned with the technical aspect of the study:

  • the authors could add a brief description as to how the sperm count, motility and morhology were assessed prior to further processing of the samples.
  • the lab equipment and chemicals lack a full description of the manufacturer, e.g. the name of the company, city and state.
  • although this study was not primarily concerned with a comparison of PIWI-LIKE 1 and 2 transcripts between patients with normal and abnormal semen parameters, however this aspect is intriguing to me. Was this comparative analysis done previously? If so, the authors could mention this in a separate paragraph in the Discussion section. This information could also demonstrate that IVF/ICSI success may not necessarily dependent on the initial semen quality. If this was not done, it could serve as a future prospect of the study.
  • a critical review of possible limitations of the study could be added to the Discussion.

Author Response

Firstly, we want to thank the reviewer for the thorough revision of our work as well as the many valuable comments and additional ideas for further analyses. We have included revisions to the manuscript according to the comments as detailed below.

My questions and/or queries are primarily concenrned with the technical aspect of the study:

Q1) the authors could add a brief description as to how the sperm count, motility and morhology were assessed prior to further processing of the samples.

A1) We introduced a respective description in l.270  - 272: “After collection, the ejaculate was stored 30 minutes at 37°C for liquefaction. Sperm con-centration and motility were assessed as described in the WHO manual, 5th edition, with at least 2x200 spermatozoa counted for each parameter.”

Q2) the lab equipment and chemicals lack a full description of the manufacturer, e.g. the name of the company, city and state.

A2) We added manufacturer’s names and affiliations.

Q3) although this study was not primarily concerned with a comparison of PIWI-LIKE 1 and 2 transcripts between patients with normal and abnormal semen parameters, however this aspect is intriguing to me. Was this comparative analysis done previously? If so, the authors could mention this in a separate paragraph in the Discussion section. This information could also demonstrate that IVF/ICSI success may not necessarily dependent on the initial semen quality. If this was not done, it could serve as a future prospect of the study.

A3) We thank the reviewer for the suggestion. We performed Mann-Whitney U tests on our data, comparing the PIWI-LIKE 1-4 transcript levels between samples with normal and abnormal semen values. In whole cohort, only PIWI-LIKE 4 expression showed a significant association to the occurrence of abnormal semen values (p = 0.023), mainly based on a higher expression in patients with subnormal ranges of morphological normal spermatozoa (p = 0.005). When analyzing the subcohorts of patients with a IVF or a ICSI treatment independently, in the IVF patients PIWI-LIKE 2 mRNA expression was significantly associated with the occurrence of abnormal sperm values (p = 0.029). However, it should be noted that only 10 (7.0%) of the patients in this group exhibited mild fertility disorders. On the other hand, in the ICSI group no association between semen values and spermatozoal PIWI-LIKE transcript levels could be shown. We added a supplementary figure to illustrate the analyses.

Q4) a critical review of possible limitations of the study could be added to the Discussion.

A4) Many thanks for raising that issue. Indeed, demonstrating the contributions of sperm RNAs to successful fertilization is only one aspect. We added the following statement: “A limitation of our study is that we do not have insight into all maternal factors involved in the fertilization process. Fertilization is a multifactorial process that involves maternal and paternal interplay. Success and failure of oocyte fertilization cannot be referred solely to one single gene family in men but remain multi-factorial.

Furthermore, evaluation of mRNA in spermatozoa as a predictor for male fertilization capacity is still experimental. The correlation between RT-qPCR data and fertilization success does not exactly reflect the mRNA content of the single spermatozoon that induces fertilization of one particular oocyte.”

Round 2

Reviewer 1 Report

Authors addressed all the concerns, but is present an accuracy on the text of the new manuscript at figures 1,2,3. Then the manuscript still need of a minor revision.

Author Response

Q1) Authors addressed all the concerns, but is present an accuracy on the text of the new manuscript at figures 1,2,3. Then the manuscript still need of a minor revision.

A1) We once again thank the reviewer for the many insightful remarks. We tried to enhance the quality of figure 1-3 once again and replaced erroneously induced commas by dots. Thus, we hope that the changes made in the figures meet the reviewer’s expectation. Best regards. Thomas Greither (on behalf of the authors)

Reviewer 2 Report

The Q7 has not been answered.

Author Response

Q1) The Q7 has not been answered. [Q7) Discussion: Line 171-177: It is necessary to discuss and justify the results on PIWI-LIKE2 expression between swim-up fraction and cell pellet.]

A1) We apologize for the mistake. In our previous work, we analyzed PIWI-LIKE gene expression in the native ejaculate, while in this study, we measured the PIWI-LIKE genes in the purified swim-up fraction. Therefore, it is hard to compare the PIWI-LIKE 2 transcript levels between our previous study and the present one. Unfortunately, it was not possible to keep an additional aliquot of native ejaculate in this study, as the samples were intended for ART treatment, and it was in the best interest of the patients to use the whole ejaculate volume for sperm purification.

However, given from the known expression pattern of PIWI-LIKE 2 ranging only up to the early pachytene spermatocytes (reviewed in Bak et al.,Clin Exp Reprod Med 2011), we speculate that there might be two separate check points: PIWI-LIKE 2 induction coupled with PIWI-LIKE 4 repression in the early stage of spermatogenesis, and PIWI-LIKE 2 repression coupled with PIWI-LIKE 1 induction in the late stage of spermatogenesis and the transition to spermiogenesis. We know that PIWI-LIKE 2 is (among other putative mechanisms) repressed by methylation of its promoter region (Giebler et al., Front Genet 2018).Therefore, one may hypothesize that PIWI-LIKE 2 may serve as surrogate marker of epigenetic maturation of a spermatozoon. However, this hypothesis warrants experimental validation, therefore we decided to not introduce it in the present manuscript.